# Synthesis of Mo_2_C by Thermal Decomposition of Molybdenum Blue Nanoparticles

**DOI:** 10.3390/nano10102053

**Published:** 2020-10-16

**Authors:** Natalia Gavrilova, Victor Dyakonov, Maria Myachina, Victor Nazarov, Valery Skudin

**Affiliations:** 1Department of Colloid Chemistry, Faculty of Natural Sciences, D. Mendeleev University of Chemical Technology of Russia, Miusskaya sq., 9, 125047 Moscow, Russia; mmyachina@muctr.ru (M.M.); nazarov@muctr.ru (V.N.); 2JSC “Kompozit”, Pionerskaya str. 4, 141070 Moscow, Russia; v.dyakonov@mail.ru; 3Department of Chemical Technology of Carbon Materials, Faculty of Petroleum Chemistry and Polymers, D. Mendeleev University of Chemical Technology of Russia, Miusskaya sq., 9, 125047 Moscow, Russia; skudin@muctr.ru

**Keywords:** molybdenum carbide, molybdenum blue, molybdenum oxide nanocluster, sol-gel method, thermal decomposition

## Abstract

In recent years, the development of methods for the synthesis of Mo_2_C for catalytic application has become especially important. In this work a series of Mo_2_C samples was synthesized by thermal decomposition of molybdenum blue xerogels obtained using ascorbic acid. The influence of the molar ratio reducing agent/Mo [R]/[Mo] on morphology, phase composition and characteristics of the porous structure of Mo_2_C has been established. The developed synthesis method allows the synthesis to be carried out in an inert atmosphere and does not require a carburization step. The resulting molybdenum carbide has a mesoporous structure with a narrow pore size distribution and a predominant pore size of 4 nm.

## 1. Introduction

Molybdenum carbides are refractory infusible compounds with high thermal and electrical conductivity, mechanical strength, and thermal stability [1]. Consideration of carbides as promising catalysts began with the pioneering work of Levi and Boudart [2], in which the high activity of tungsten carbide in the isomerization of 2,2-dimethylpropane was shown. It was noted that the activity of tungsten carbide was comparable to that of a platinum catalyst.

The arisen interest in these compounds has intensified research on transition metal carbides as catalysts for various reactions. In particular, the catalytic activity of molybdenum carbide in the reactions of hydrazine hydrate decomposition [3], cellulose conversion [4], and hydrogenation [5,6] was noted.

High activity of molybdenum carbides was also observed in gas-phase processes, most of which are reactions involving hydrogen: reforming of hydrocarbons [7,8,9,10,11,12], selective oxidation [13], water gas shift reaction [14,15,16], desulfurization [17,18], aromatization [19], isomerization [20,21], selective reduction of CO_2_ [22], as well as biomass conversion [23,24]. Such a similar behavior of molybdenum carbides with platinum group metals makes it possible to explain the structural features of carbides [25,26,27,28,29].

The metal lattice of molybdenum carbides can be represented by face-centered cubic, simple hexagonal, or hexagonal with close packing [30,31]. Three types of bonds are present in carbides: metallic (Me–Me), covalent (Me–C), and ionic (Me–C) [32,33]. Due to the inserted carbon atom into the metal lattice, the distance between the metal atoms increases, which leads to an increase of the Mo d-bands density of states.

It is also known that hybridization of carbon and molybdenum orbitals leads to a wider energy distribution in molybdenum carbide in comparison with the metal matrix. It is due to these electronic properties that molybdenum carbides exhibit catalytic activity similar to that of Pt in various reactions [34].

Another important advantage of molybdenum carbides is their resistance to coking and the resistant to many catalytic poisons [35,36,37]. The unique physicochemical properties of group VI metal carbides, as well as their lower cost, mechanical strength, and higher electrical conductivity, make it possible to consider these compounds as an alternative to catalysts based on platinum group metals.

Such results lead to the necessity to develop methods for the synthesis of bulk and supported catalysts based on molybdenum carbides. It should be noted that these carbides are not common in nature and can only be obtained synthetically.

The traditional method for the synthesis of carbides is a metallurgical method based on long-term temperature treatment of metal and carbon [38]. Due to the high synthesis temperature (1200–1500 °C), carbides have an extremely low specific surface area which limits their use in catalysis.

There are works devoted to the mechanochemical and electrochemical method for the synthesis of molybdenum and tungsten carbides [39,40]. However, the most widely used method is temperature-programed reduction (TPR), which is based on gas-solid reactions. Trioxides MoO_3_ [41,42] less often dioxides MoO_2_, are used as sources of molybdenum, and a mixture of hydrocarbon gases (CH_4_, C_2_H_6_, C_3_H_8_, C_4_H_10_) with H_2_ or aromatic compounds [43,44,45,46].

A common disadvantage of methods based on TPR is the deposition of carbon on the surface of Mo_2_C particles in the form of a polymer film (the thickness of which can reach 12 nm), which not only leads to a decrease in the specific surface area, but also blocks catalytically active sites [47]. It should also be noted that there is a need to provide increased safety measures when using H_2_/C_n_H_2n+2_ mixtures in the synthesis, which limits the use of this method on an industrial scale.

The development of methods for producing not only bulk, but also supported catalysts is especially actual. For these purposes, the most suitable are methods based on solid–liquid reactions. The synthesis methods can be attributed to reduction reactions, as a result of which molybdenum carbide is formed, or to liquid-phase synthesis with the participation of soluble organic compounds, which serve as a carbon source during further heat treatment. But at present, this group of methods is less common then solid-solid or gas-solid reactions.

In [48,49] urea was used as a carbon source. Molybdenum pentachloride was mixed with ethyl alcohol and urea, and the reaction resulted in the formation of a gel, the heat treatment of which in an inert medium led to the formation of α-Mo_2_C nanoparticles.

In [50], a solution of sucrose (glycine, alanine) was used, which was added with constant stirring to a solution of ammonium heptamolybdate, after which the resulting gel was dried, and heat-treated. It was shown that the formation of molybdenum carbides occurs at low temperatures. In this case, depending on the amount of sucrose, the formation of various modifications of Mo_2_C was observed.

An interesting way to synthesize molybdenum carbide is to use molybdenum blue as precursor. Molybdenum blue is a compound containing polyoxometalates (POMs)—giant molybdenum oxide clusters [51]. Molybdenum blue clusters exhibit unique physicochemical properties due to the feature of their structure. Clusters are formed by self-assembly of the initial molybdenum oxide blocks, which makes it possible to change the synthesis conditions and obtain structures with the required composition (Mo_138_, Mo_154_, Mo_368_ etc.), shape (spherical, toroidal, lemon-type), and size (1–8 nm) of clusters [52,53].

Currently, molybdenum blue is being actively studied and explored as objects for production of hybrid materials, drug delivery systems, nanoreactors and membranes [54]. Due to their nanoscale and monodispersity, molybdenum blues can serve as excellent precursors for the synthesis of various catalytic systems.

Molybdenum blues are stable dispersions of nanocluster molybdenum compounds; they can be obtained by reduction of molybdates in an acidic medium. When organic reducing agents are used, the reducing agent is also a source of carbon in the synthesis of molybdenum carbide. In this case, the introduction of additional carbon sources, not in the liquid phase, not in the gas phase, is not required, this avoids carrying out the stage of temperature programed carburization.

In [55,56] the synthesis of Mo_2_C by thermal decomposition of molybdenum blue, obtained by reduction of glucose, hydroquinone was described. As a result of thermal decomposition, molybdenum carbide Mo_2_C and carbon were formed. Since the synthesis of stable dispersions required a significant excess of the reducing agent ([R]/[Mo] = 7 for glucose and 4 for hydroquinone). In [55] the successful use of dispersions for the preparation of membrane catalysts based on Mo_2_C was demonstrated.

It is known that ascorbic acid is a stronger reducing agent and allows the synthesis of molybdenum blues without such a large excess of the reducing agent. In [57] was reported that stable dispersions of molybdenum blue were synthesized in the range of molar ratio [R]/[Mo] from 0.2 to 5.0 In this work, samples of molybdenum carbide were synthesized, and the effect of the [R]/[Mo] molar ratio on its main characteristics, including morphology, phase composition, and porous structure, was revealed.

## 2. Materials and Methods

### 2.1. Materials

Ammonium heptamolybdate (NH_4_)_6_Mo_7_O_24_∙4H_2_O, ascorbic acid C_6_H_8_O_6_, and hydrochloric acid HCl were purchased in CT Lantan, Moscow, Russia. All chemicals were used without further purification.

### 2.2. Synthesis of Molybdenum Blue Dispersion

Molybdenum blue were synthesized by reducing a solution of ammonium heptamolybdate (0.07 M, calculated on monomolybdate) by ascorbic acid in the presence of hydrochloric acid. The synthesis was carried out at room temperature and the following conditions: molar ratios [R]/[Mo] = 0.2–5.0, molar ratios [H]/[Mo] = 0.8. The details of the synthesis can be found in [57]. Aggregative stable dispersions can be synthesized with a rather wide range of molar ratios [R]/[Mo] = 0.4–5.0. Which makes it possible to use these systems to obtain molybdenum carbide without additions of another carbon source.

### 2.3. Molybdenum Blue Dispersion Characterization

Particle size distribution of the molybdenum blue dispersion was measured using the dynamic light scattering (DLS) technique. The measurements were performed with a Photocor Compact-Z instrument (Photocor LLC, Moscow, Russia) at a wavelength of 658 nm.

The particle size of the molybdenum blue was studied using a LEO 912AB Omega (Carl Zeiss, SMT AG Oberckochen, Germany) transmission electron microscope. Images were acquired at 100 kV accelerating voltage. The analysis of the microphotographs and the calculation of particle sizes were carried out using the Image Tool V.3.00 (Image Tool Software, UTHSCSA, San Antonio, TX, USA).

X-ray photoelectron spectroscopy (XPS) spectra were recorded using a ESCA + X-ray photoelectron spectrometer (OMICRON Nanotechnology GmbH, Taunusstein, Germany).

### 2.4. Procedure of Synthesis of Mo_2_C

Xerogels of molybdenum blues with different molar ratio [R]/[Mo] were prepared by drying the dispersion at room temperature. Molybdenum carbide was obtained by calcining xerogels of molybdenum blue in an atmosphere (N_2_) at 900 °C.

### 2.5. Mo_2_C Characterization

The molybdenum blue xerogels were characterized by thermal analysis using an SDT Q600 thermal analyzer (TA Instruments, New Castle, DE, USA). Xerogel samples (10 mg) were calcined in ceramic crucibles (Al_2_O_3_) in inert atmosphere (Ar) in the temperature range from 25 to 1000 °C at a heating rate of 5 °C/min.

X-ray diffraction (XRD) patterns were recorded at room temperature over the scanning range (2θ) of 20–90° with a step 0.020° and scan step of 5°min^−1^ using a D/MAX 2500 diffractometer (Rigaku Corporation, Tokyo, Japan) with CuKα radiation. Accelerating voltage was 50 kV, cathode filament current 250 mA (total X-ray tube power—12.5 kW). The phases present in the samples were identified using JCPDS Powder Diffraction File data.

The samples morphology was studied using a JSM 6510 (Jeol, Tokyo, Japan) scanning electron microscope with an EDX + SSD X-MAX X-ray microanalysis system. Before measurement palladium layer on the samples by ion sputtering was deposited. Size distribution were obtained from the measurement of at least 500 random selected particles per sample.

The specific surface area, pore volume and pore distribution were calculated on the basis of data of N_2_ low-temperature adsorption on a Gemini VII (Micromeritics, Norcross, GA, USA) surface area and porosity analyzer at the Mendeleev University Center for Shared Use of Equipment. The specific surface area of the samples (S_a_) was calculated using the BET method. The total specific pore volume (ΣV) was determined at a maximum relative pressure of 0.995. The mesopore volume (V_meso_), the mesopore size distribution was calculated by the BJH method. The micropore volume (V_0_) was determined based on the Dubinin–Radushkevich equation. The presence of micropores in samples, the true value of micropore volume (V_t_) was calculated using the t-plot de Boer method.

### 2.6. Catalyts Activity Test

The study of the catalytic activity of Mo_2_C powder samples was carried out in the reaction of carbon dioxide conversion of methane. The reaction was carried out in a quartz reactor with a fixed catalyst bed. The volume of the powdered catalyst was up to 1 sm^3^. The powder was diluted with quartz and placed in the isothermal zone of the reactor.

The volumetric velocity of the gas mixture (CH_4_/CO_2_ = 1/1) varied in the range from 30 to 320 sm^3^/min. The flow rate and composition of methane and carbon dioxide were maintained by using mass flow controllers EL-FLOW (“Bronkhorst High Tech”, AK Ruurlo, The Netherlands). The temperature regime of the catalytic process was set using a ThermoDat-17E6 temperature controller (“Control Systems”, Perm, Russia). The flow rate of the product mixture was determined with an ADM G6691A flow meter (“Agilent Tech.”, Santa Clara, CA, USA). The reaction was carried out in the temperature range of 600–930 °C. Temperature control was carried out using chromel–alumel thermocouples.

A quantitative analysis of the reaction mixture composition was carried out by using a gas chromatograph Crystall 5000 (CJSC SKB “Chromatec”, Yoshkar-Ola, Russia) equipped with two thermal conductivity detectors and chromatographic columns (HayeSep R 80/100, NaX 60/80).

To compare the catalytic activity of the samples, the rate constants (k) of methane dissociation was calculated, which was accepted as the limiting stage in the carbon dioxide conversion of methane. The specific constant rate (k_s_) was calculated by dividing the constant k by the mass and surface area of the catalyst.

## 3. Results

### 3.1. Molybdenum Blue Characterization

Solid–liquid methods for the synthesis of molybdenum carbides are of interest because the use of a reducing agent or stabilizer in the synthesis of dispersions allows them to be used simultaneously as a carbon source. In this case, the subsequent temperature programed carburization step can be excluded. The possibility of varying the synthesis conditions, in particular the amount of organic compound, allows the use of molybdenum carbide with the absence or with the required amount of free carbon. This idea is used in this work.

For the synthesis of molybdenum carbide, a dispersion of molybdenum blue was used. Molybdenum blues were obtained according to the reduction of molybdate in an acidic medium. Ascorbic acid was used as a reducing agent.

Molybdenum blues are dispersions of molybdenum oxide clusters—giant polyoxometalate structures. In Figure 1a shows micrographs of molybdenum blue particles synthesized using ascorbic acid.

The predominant particle diameter, determined from the analysis of micrographs, is 3 nm (Figure 1b). According to dynamic light scattering data, the hydrodynamic diameter of molybdenum blue particles is 3.4 nm (see Figure 1c). This is in good agreement with the data on the size of molybdenum blue clusters [51]. It was found that under the selected synthesis conditions, the dispersed phase of molybdenum blues is represented by toroidal clusters [57]. It is interesting to note that regardless of the molar ratio [R]/[Mo], the size and shape of the molybdenum blue particles do not change.

The synthesized dispersions are chemically and aggregative stable in the pH range from 0.8 to 3.0. At pH values below 0.8, particle coagulation is observed with the formation of a precipitate; at pH values above 3.0, toroidal clusters dissolve with the formation of different dissolved forms of molybdates. In the pH range of aggregate stability, molybdenum blue can be concentrated without loss of sedimentation stability up to a concentration of 8 wt.%, above which gel formation occurs. The sol-gel transition makes it possible to use these dispersed systems not only for obtaining bulk catalysts, but also for supported catalysts and catalytic layers of membranes.

A distinctive feature of molybdenum blue is that in the composition of polyoxomolybdate complexes, molybdenum is present in the form of Mo^V^ and Mo^VI^. To confirm the presence of reduced molybdenum Mo^V^ in the analyzed samples, XPS spectroscopy was used. An overview spectrum of molybdenum blue particles is shown in Figure 2. The energy range of survey XPS spectra is usually divided into three characteristic parts. The first part is a region with a low electron binding energy (range from 0 to 15 eV) associated with the electrons of the outer valence orbitals. The range from 15 to 50 eV corresponds to the energies of the electrons of the internal valence orbitals. In the third region of the spectrum (50 eV and more), lines correspond to the energies of internal electrons [58].

According to the spectrum presented, the elemental composition of molybdenum blues is represented by molybdenum, oxygen, carbon, as well as the impurity content of potassium and chlorine (KCl was used to isolate particles from dispersions).

To establish the oxidation state of an element, the lines of internal electrons are usually used; for molybdenum, this is the region of the line of Mo3d electrons. This region of the spectrum is shown in Figure 2b (black line—experimental data; red line—fit sum, yellow and green lines—spectrum interpretation). Molybdenum in molybdenum oxide clusters synthesized in the presence of ascorbic acid is presented in two forms: Mo^6+^ d 5/2 and Mo^6+^ d 3/2 and Mo^5+^ d 5/2 and Mo^5+^ d 3/2.

Thus, XPS spectroscopy confirms the presence of reduced molybdenum Mo^5+^ in the composition of molybdenum oxide clusters. From the assessment it follows that the content of reduced molybdenum Mo^5+^ in molybdenum blues synthesized using ascorbic acid is 29%. The results obtained agree with the literature data on the degree of molybdenum reduction in toroidal clusters [59].

### 3.2. Mo_2_C Preparation

Under synthesis conditions, thermal decomposition of molybdenum blues can lead to the formation of molybdenum carbide. The source of molybdenum is molybdenum clusters, and the source of carbon is ascorbic acid and its oxidation products. In this case, thermal decomposition must be carried out in an inert atmosphere.

To determine the temperature of formation of molybdenum carbides, a thermal analysis of a xerogel of molybdenum blue was carried out (as an example, Figure 3 shows the results of TGA for a sample of molybdenum blue xerogel synthesized with molar ratio [R]/[Mo] = 1.0). Significant weight loss occurs in the temperature range 50–250 °C (30% of the initial sample weight). In the temperature range from the initial value to 100 °C, the evaporation of free water retained in the interparticle space occurs. This is indicated by the negative value of the heat effect in this interval. When the temperature rises to 250 °C bound water is released. The fact that particles of molybdenum blue are highly hydrated is known in the literature [51] and the data obtained are in good agreement with this.

Thermolysis processes occur in the temperature range from 250 to 700 °C. The decomposition of ammonium chloride (presented in the disperse medium of molybdenum blues) is observed at temperature range of 350–400 °C. It is known that thermolysis for organic substances generally ends at 600–650 °C. The thermal effects of such reactions are mostly positive. In the presented DTA curve, positive values of the heat effect are also observed in this interval. In this case, products are released into the gas phase. The analysis of these products was carried out by gas chromatography. It was found that they consist of CO_2_, H_2_ and CH_4_, which are the products of thermolysis of ascorbic acid.

At temperatures of 700–750 °C there is a sharp change in the heat effect of the reaction, the sample weight and the rate of its decrease, which correspond to the maximum on the DTG curve. Usually, the negative effect of the reaction is attributed to phase transformations or to reactions of formation of substances, which is also accompanied by a change in the sample mass. These changes may indicate the formation of molybdenum carbide.

To obtain information on the phase composition of the material, X-ray diffraction patterns were obtained for samples after heat treatment in an N_2_ medium at different temperature. The results of these studies are shown in Figure 4.

As can be seen at diffraction patterns, no reflections of crystalline phases for samples calcined at 500 and 500 °C are observed, which is in good agreement with the TGA results. An increase in temperature leads to the appearance of reflections of molybdenum carbides: β-Mo_2_C [08–0387] and η-MoC [08–0384]. The formation of carbides is completed at a temperature of 800 °C, and further heat treatment does not lead to a change in the phase composition of the samples.

The influence of the molar ratio [R]/[Mo] on the phase composition of molybdenum carbides formed from molybdenum blue was investigated. The research results are shown in Figure 5. As can be seen from the diffractograms the sample synthesized at molar ration [R]/[Mo] = 0.2 is MoO_2_, i.e., this amount of reducing agent is insufficient to form molybdenum carbide. Samples syntesized with [R]/[Mo] = 0.4–0.8 are characterized by the presence of one phase of molybdenum carbide—β-Mo_2_C [08–0387]. The sample with [R]/[Mo] = 0.8 is characterized by the maximum intensity of phase reflections and the absence of free carbon reflections. This indicates the completeness of the reaction of the molybdenum carbide formation. An increase in the content of the reducing agent leads to the appearance of η-MoC [08–0384] and an increase in free amorphous carbon, this becomes especially noticeable when the ratios [R]/[Mo] ≥ 2 (Figure 5b).

The conducted research has established the path of formation of molybdenum carbide during the thermal decomposition of molybdenum blue xerogels. it was shown that molybdenum carbide is formed at a relatively low temperature (750–800 °C) by thermal action on xerogels in an inert environment.

### 3.3. Mo_2_C Characterization

Figure 6 shows micrographs of molybdenum carbide samples obtained by heat treatment in inert atmosphere of xerogels synthesized using ascorbic acid. Xerogels of molybdenum blue synthesized with a low content of the reducing agent ([R]/[Mo] = 0.6–1.0) are aggregates of primary crystalline particles of Mo_2_C. The aggregates are up to 200 microns in size; their shape resembles fragments with clear edges and chips. No other types of particles were observed in the samples. The average size (number average diameter) of the primary particles is of 300–400 nm. Under such synthesis conditions, molybdenum carbide is formed with a corpuscular porous structure, where the pores are the spaces between the sintered particles.

An increase in the content of the reducing agent ([R]/[Mo] ≥ 2) leads to the appearance of carbon particles with smooth surface, in the matrix of which particles of molybdenum carbide are present. The particle size of molybdenum carbide does not change with increasing [R]/[Mo] and is about 400 nm. Under these synthesis conditions, a bidisperse porous structure is presents: a spongy structure of free carbon formed by thermal decomposition of organic precursor and a corpuscular porous structure of molybdenum carbide.

Molybdenum blues can be synthesized by ascorbic acid reduction in a fairly wide range of molar ratios (from insufficient carbon content to obtain molybdenum carbide to its excess). Below are the results in which it is possible to trace the influence of the molar ratio on the main characteristics of molybdenum carbide including porous structure. Figure 7a,b shows nitrogen adsorption isotherms at molybdenum carbide samples at 77K.

As can be seen from the presented adsorption data, an increase in [R]/[Mo] leads to a change of the form of isotherms. According to Brunauers’ classification, all isotherms are of type IV. However, increase in the [R]/[Mo] ratio leads to a sharp increase in adsorption. In the region of low pressures, a sharp increase of adsorption appears, indicating the presence of micropores. This is especially pronounced for the samples synthesized at [R]/[Mo] ≥ 2.0.

To confirm the presence of micropores in the samples, the De Boer t-method was used (see Figure 6c,d). The samples synthesized at [R]/[Mo] ≤ 1.0 are mesoporous materials. With an increase in the content of the reducing agent in the samples, microporosity appears. The results obtained are in good agreement with the data of X-ray diffraction analysis and electron microscopy, according to which the synthesis of molybdenum carbide with a ratio [R]/[Mo] ≥ 2.0 leads to the formation of a second phase in the material, free carbon, which makes the main contribution to the microporosity of the synthesized samples.

The size distribution of meso- and micropores is rather narrow (see Figure 8), the prevailing diameters (width of the slit pore) of Mo_2_C and amorphous C are 4.0 and 1.0 nm, respectively.

Changes in the porous characteristics of samples of this series can be seen in Table 1. Molybdenum carbide synthesized without excess of the reducing agent has a specific surface area of 1.4–3.2 m^2^/g. However, it should be emphasized that the synthesized samples have a mesoporous structure with a very narrow pore size distribution and a predominant diameter of 4 nm.

An increase in the content of the reducing agent leads to the formation of free carbon and the appearance of a microporous structure with a predominant pore size of 1 nm, which contributes to the specific surface area of the synthesized materials.

The low value of the specific surface area is due to the fact that sintering is inevitable in the process of obtaining powdery materials by heat treatment. This disadvantage can be avoided by preparing supported catalysts on porous supports.

### 3.4. Mo_2_C Catalytic Activity

The catalytic activity of the synthesized samples was investigated in the reaction of methane carbon dioxide conversion. In the studies, we used Mo_2_C samples obtained from molybdenum blue xerogels with different ratio [R]/[Mo] = 0.6; 1.0; 2.0; 5.0. The main reaction products are H_2_, CO, and H_2_O. No other reaction products were found. The degree of conversion of the starting reagents depends on both the reaction temperature and the contact time. As an example, Figure 9 shows the dependences of the degree of conversion of methane and carbon dioxide on the temperature of the reaction on one of a series of samples. As can be seen from the above figure, the conversion of methane begins at a temperature of 650 °C and reaches its maximum value at 850 °C. Under reaction conditions, the molar ratio H_2_/CO reaches 0.65 (see Figure 9b) and can be increased by changing the reaction conditions.

It was found that the catalytic activity of the samples depends on the molar ratio [R]/[Mo] used in the synthesis of molybdenum blues. For a correct comparison of the catalytic activity of the samples, a kinetic study was carried out. Which made it possible to calculate the rate constant of the methane dissociation for each of the Mo_2_C samples. Table 2 shows the results obtained. The table also shows the specific rate constants (k_s_) per unit surface area of the catalyst.

## 4. Discussion

The conditions for obtaining molybdenum carbide by thermal decomposition of molybdenum blue xerogels were determined. It was shown that molybdenum carbide is formed at a relatively low temperature (750–800 °C) by thermal decomposition of xerogels in an inert environment. It was found that the formation of carbide occurs due to the interaction of molybdenum dioxide and carbon, which are formed as a result of thermal action on xerogels of molybdenum blue.

The formation of molybdenum carbide from xerogels leads to the formation of a bidispersity porous structure with developed micro- and mesoporosity. Mesopores are formed during sintering of molybdenum carbide particles; microporosity is formed during the formation of free carbon, which is formed when there is an excess of reducing agent in molybdenum blue. The pore size distribution of meso- and micro-pores is rather narrow.

The specific surface area is not very high and comparable to the samples obtained by other methods [60,61]. However, when molybdenum carbide is applied to various supports, the specific surface area can be significantly increased. Molybdenum blue can be used to impregnate porous supports using sol-gel technology. We have achieved similar results in our previous works, where dispersions of molybdenum blue synthesized with other organic reducing agents were used in the preparation of supported catalysts [55].

A distinctive feature of this method is that in the samples obtained with an excess of the reductant ([R]/[Mo] ≥ 2.0), the formed carbon does not cover the molybdenum carbide as film, but is a matrix in which the molybdenum carbide particles are located. Such carbon–carbide samples may be of interest as catalysts for desulfurization and different hydroprocesses.

The synthesized samples of Mo_2_C showed high catalytic activity in the reaction of carbon dioxide conversion of methane. It was found that 100% methane conversion is achieved at 850 °C. This is in good agreement with the literature data on the activity of molybdenum carbide in this reaction [11,62]. The experiments carried out made it possible to establish that the highest catalytic activity is exhibited by carbide samples obtained by heat treatment of molybdenum blue xerogels with a low molar ratio [R]/[Mo] = 0.6; 1.0. The best results are observed for a sample with a phase composition of β-Mo_2_C, η-MoC.

## 5. Conclusions

A series of molybdenum carbide was successfully synthesized by a thermal decomposition of molybdenum blue xerogels. The developed synthesis method makes it possible to obtain molybdenum carbide at a relatively low temperature without using a carburization step. The molar ratio [R]/[Mo] used in the synthesis of molybdenum blues is a parameter that has a significant effect on the properties of molybdenum carbide. By varying [R]/[Mo], it is possible to synthesize various modifications of molybdenum carbides—the β-Мo_2_С and η-МoС.

The synthesized samples of molybdenum carbides exhibit high activity in the reaction of carbon dioxide conversion of methane. The developed method based on the use of nanoparticles of molybdenum blue makes it possible to use all the advantages of the sol-gel method in the preparation of catalytic materials and can be used to obtain massive, supported catalysts and catalytic membranes.

## Figures and Tables

**Figure 1 nanomaterials-10-02053-f001:**
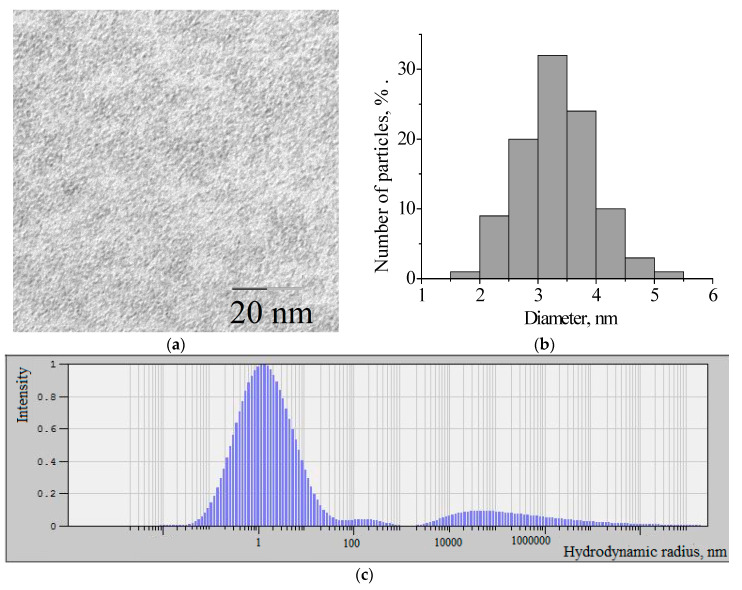
(**a**) TEM image of molybdenum blue particles; (**b**) particles distribution histogram of molybdenum blue; (**c**) hydrodynamic size distribution of molybdenum blues particles.

**Figure 2 nanomaterials-10-02053-f002:**
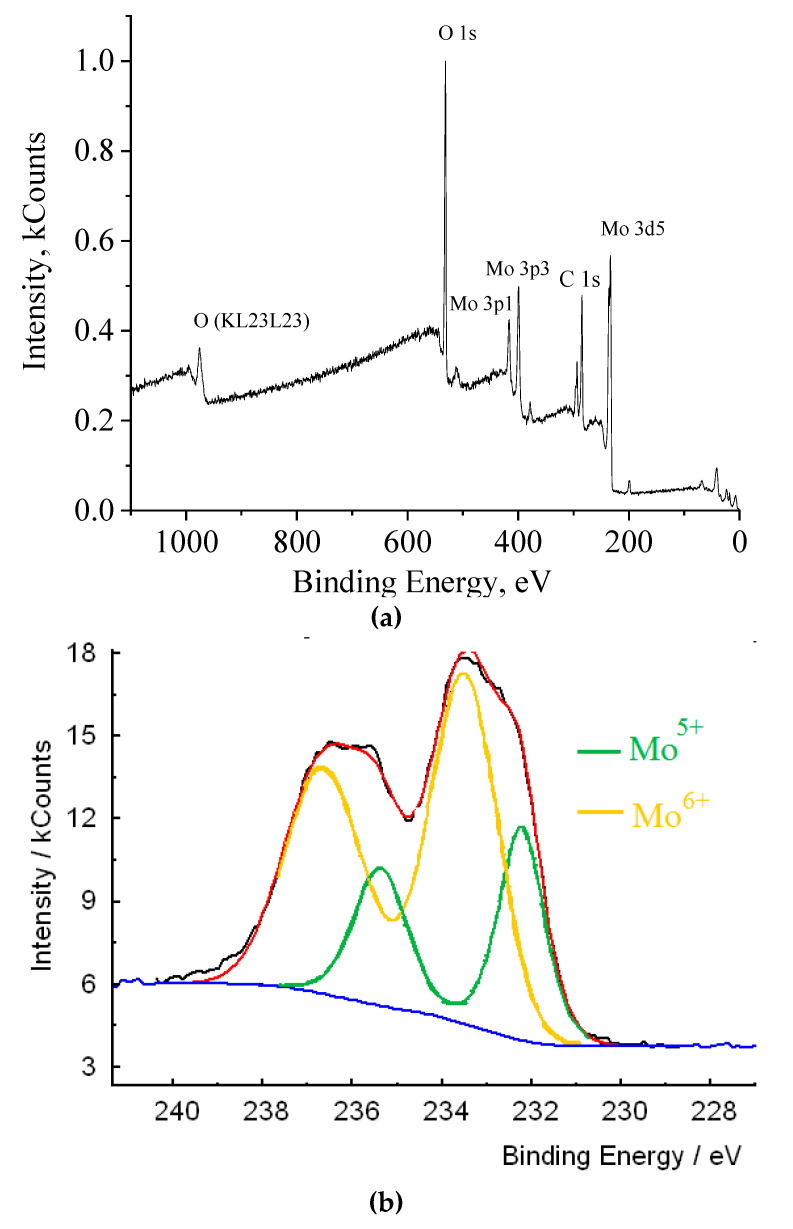
X-ray photoelectron spectroscopy (XPS) spectrum of molybdenum clusters: (**a**) survey spectra; (**b**) Mo3d spectra (black line—experimental data; red line—fit sum, yellow and green lines—spectrum interpretation).

**Figure 3 nanomaterials-10-02053-f003:**
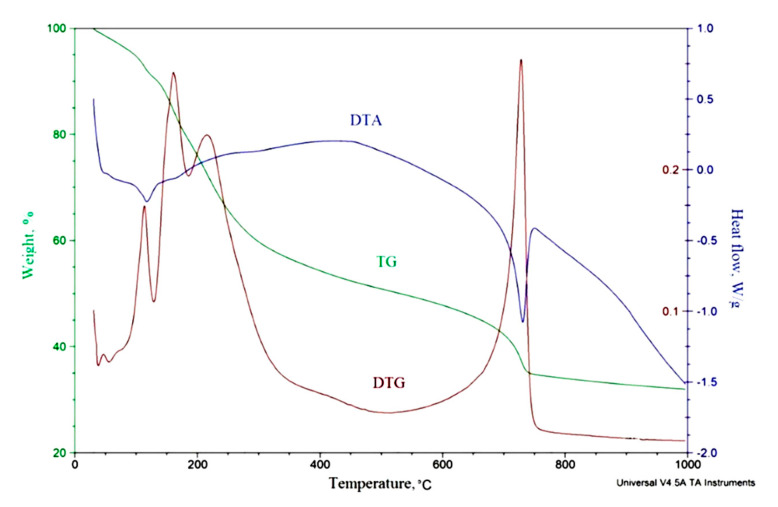
DTA curves of molybdenum blue xerogel synthesized with molar ratio [R]/[Mo] = 1.0 (inert atmosphere).

**Figure 4 nanomaterials-10-02053-f004:**
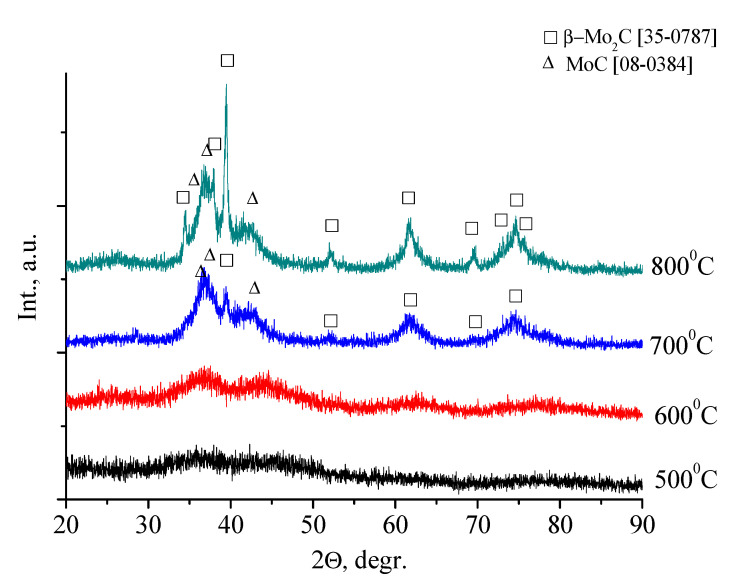
XRD pattern of molybdenum blues xerogels synthesized with molar ratio [R]/[Mo] = 1.0, calcined at different temperature in inert atmosphere (N_2_).

**Figure 5 nanomaterials-10-02053-f005:**
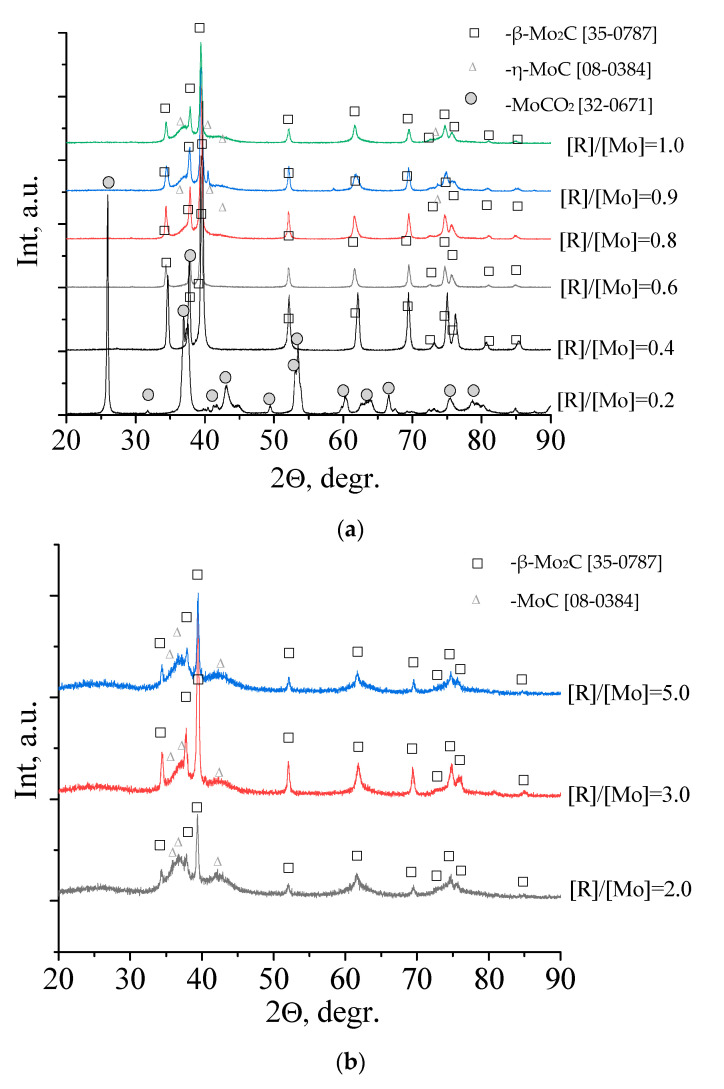
XRD pattern of molybdenum blues xerogels with different ratio [R]/[Mo], calcined at 900 °C in inert atmosphere (N_2_). (**a**) [R]/[Mo] = 0.4–1.0; (**b**) [R]/[Mo] = 2.0–5.0.

**Figure 6 nanomaterials-10-02053-f006:**
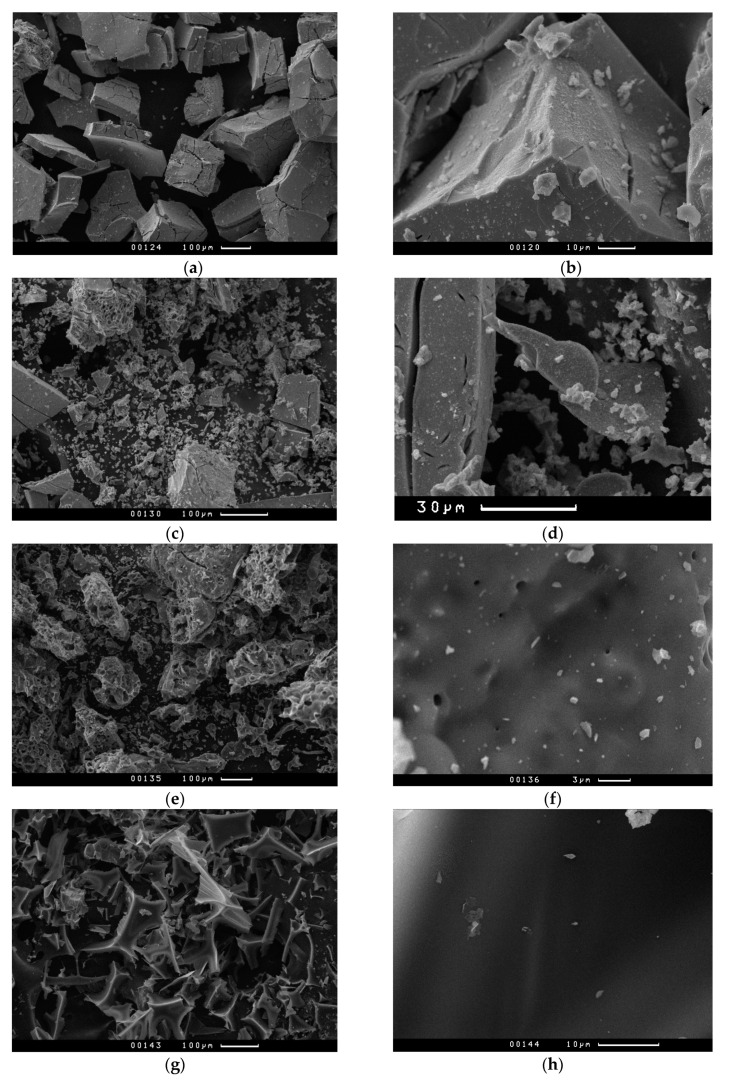
SEM images of Mo_2_C synthesized from molybdenum blue xerogels (calcined at 900 °C in inert atmosphere) with different ratio [R]/[Mo]: 0.8 (**a**,**b**), 1.0 (**c**,**d**), 2.0 (**e**,**f**), 5.0 (**g**,**h**).

**Figure 7 nanomaterials-10-02053-f007:**
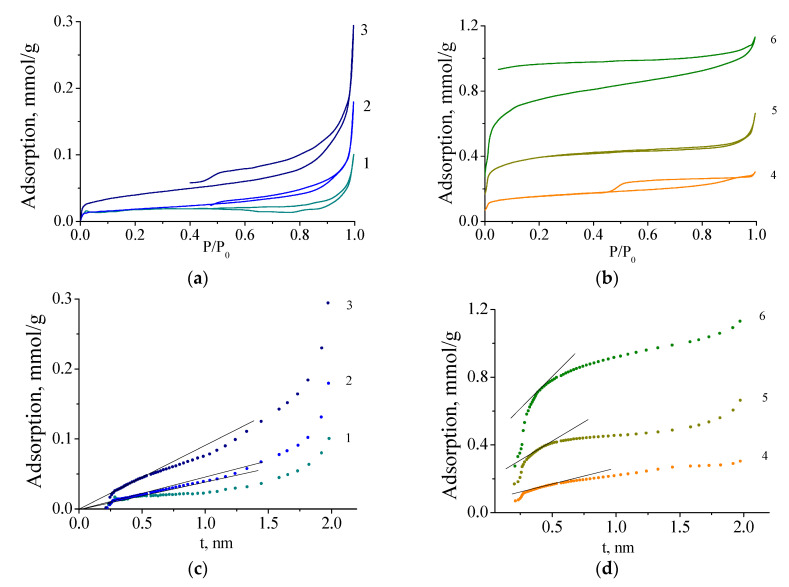
(**a**,**b**) Nitrogen adsorption isotherms and (**c**,**d**) t-plots on Mo_2_C synthesized from molybdenum blue xerogels (calcined at 900 °C in inert atmosphere) with different ratios [R]/[Mo]: 0.6 (1), 0.8 (2), 1.0 (3), 2.0 (4), 3.0 (5), 5.0 (6).

**Figure 8 nanomaterials-10-02053-f008:**
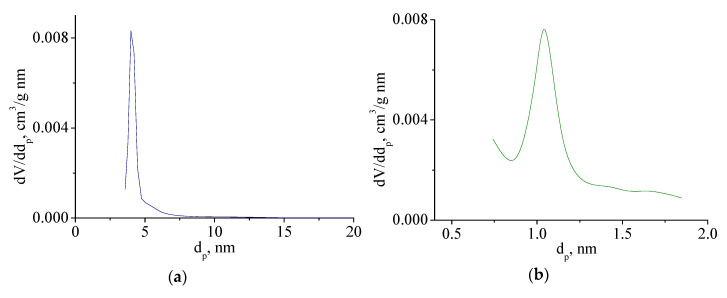
Meso (**a**) and micropore (**b**) size distribution of Mo_2_C, synthesized from molybdenum blue xerogel with ratio [R]/[Mo] = 2.0, calcined at 900 °C in inert atmosphere.

**Figure 9 nanomaterials-10-02053-f009:**
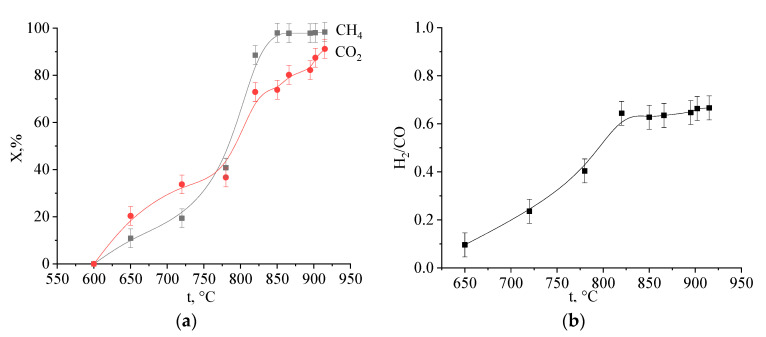
Dependences of the (**a**): substance concentrations on temperature (**b**): molar ratio H_2_/CO on temperature on Mo_2_C catalyst ([R]/[Mo] = 1.0). Flow rate of the mixture is 30 cm^3^/min.

**Table 1 nanomaterials-10-02053-t001:** Characteristic of porous structure of Mo_2_C synthesized from molybdenum blue xerogels with different ratio [R]/[Mo], calcined at 900 °C in inert atmosphere.

Molar Ratio [R]/[Mo]	Parameters						
	S_a_ (BET), m^2^/g	S_a_ (t-plot), m^2^/g	∑V, sm^3^/g	V_meso_ (BJH), sm^3^/g	V_0_ (DR), sm^3^/g	V_t_, (t-plot) sm^3^/g	d_p_, nm
0.6	1.4	1.4	0.0033	0.0023	0.0006	-	4
0.8	1.4	1.4	0.0039	0.0031	0.0008	-	4
0.9	1.4	1.4	0.0043	0.0032	0.0007	-	4
1.0	3.2	3.2	0.0099	0.0062	0.0020	-	4
2.0	12.4	7.1	0.1053	0.0067	0.0051	0.0022	4; 1
3.0	31.5	14.8	0.0227	0.0061	0.0130	0.0070	4; 1
5.0	63.0	27.7	0.0391	0.0046	0.0253	0.0137	4; 1

S_a_—specific surface area calculated using BET equation, ΣV—the total specific pore volume, Vmeso—mesopore volume, V_0_—the micropore volume determined based on the Dubinin–Radushkevich equation. V_t_—value of micropore volume calculated using the t-plot de Boer method, d_p_—predominant pore size.

**Table 2 nanomaterials-10-02053-t002:** Catalytic activity of Mo_2_C synthesized from molybdenum blue xerogels with different ratio [R]/[Mo].

Molar Ratio [R]/[Mo]	Parameters			
	Phase Composition	S_a_ (BET)*, m^2^/g	k^900^, s^−1^	k_s_^900^, s^−1^ m^−2^
0.6	β-Mo_2_C	0.6	2.64	8.00
1.0	β-Mo_2_C, η-MoC	1.4	8.20	22.44
2.0	β-Mo_2_C, η-MoC, C	1.6	0.45	0.50
5.0	β-Mo_2_C, η-MoC, C	16.0	0.60	0.03

Sa—specific surface area calculated using BET equation, k_900_—constant rate calculated at 900 °C, *—after reaction.

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
