# Peer review of "Synthesis of Mo2C by Thermal Decomposition of Molybdenum Blue Nanoparticles"

_nanomaterials, 2020, doi:10.3390/nano10102053_

Round 1

Reviewer 1 Report

The manuscript by Gavrilova and co-workers, titled “Synthesis of Nanosized Mo2C by Thermal Decomposition of Molybdenum Blue” describes the synthesis of molybdenum carbides nanoparticles starting from thermal decomposition of molybdenum blue nanoparticles, which were in turn prepared by using ammonium heptamolybdate and ascorbic acid at different molar ratio in the presence of hydrochloric acid.

Both molybdenum blue and molybdenum carbides nanoparticles were fully characterized. This work seems the prosecution of the paper “Synthesis of Molybdenum Blue Dispersions Using Ascorbic Acid as Reducing Agent”, Colloids Interfaces 2020, 4, 24, by the same authors, and, personally, I would have merged both manuscripts in one paper. Anyway, the work seems to be well done and the experimental results are scientifically soundly. References need to be updated, especially the ones describing the catalytic uses of molybdenum carbides. English needs to be polished. The work could have been improved by reporting a catalytic test of the new prepared nanomaterials. Therefore, I suggest accepting the manuscript after the above-mentioned minor revision.

Author Response

The authors are grateful to the Reviewer for such a careful and detailed reading of the manuscript. Yes, indeed, this manuscript is one of a series of planned articles on the use of molybdenum blue as a precursor for molybdenum carbide catalysts. These objects (molybdenum blue) have shown to be promising precursors of various types of catalysts; in particular, they are of interest for the preparation of supported catalysts and catalytic membranes, and positive results have already been obtained on the catalytic activity of these types of catalysts. These results will be published shortly.

The references have been updated and the manuscript has been modified accordingly. The following references have been modified: 3, 6, 11, 12, 14, 19, 23, 32, 43, 51, 52.

The English language has been improved. The corrections made are highlighted in the manuscript.

The data on the catalytic activity of the samples in the reaction of carbon dioxide conversion of methane have been added to the manuscript. Sections 2.5 (Catalyst activity test) and 3.4 (Mo2C catalytic activity) has been added and were expanded section of Discussion and Conclusions.

Reviewer 2 Report

The use of molybdenum carbide as catalyst depends on its porosity. The authors already note that the obtained porosity is relatively low when compared to other synthesis methods. I consider that first a more quantitative comparison with reported results is compulsory; the obtained porosity in this contribution should be compared to those reported in the literature. This relatively low porosity cast doubts about the benefits of using the reported method to obtain molybdenum carbide. Moreover, from Fig 5, the size of molybdenum carbide particles is far above the nanometer range. So given the fact that Nanomaterials journal scope is “involves nanomaterials, with respect to their science and application”, I consider that in its present state this contribution is not suitable for Nanomaterials journal.

Minor comments:
Thermal analysis is an indirect technique, so it is necessary the use of ancillary methods to identify the different steps. In page 7 the authors identify three stages that are assigned to water loss, decomposition of ammonium chloride and formation of molybdenum carbide, but not a single evidence of these assignments is provide. Not even, the observed mass loss is compared to the expected mass loss for such transformations. Therefore, the analysis of the TG curve is quite speculative. The authors should provide evidences of such processes or indicate that their identification is just a guess based on the expected thermal evolution of the precursor.
The authors claim that molybdenum carbide is formed in the temperature range from 750 to 850ºC, but the TG curve on fig. 3 shows that the mass is not stable above 800ºC; there is a smooth mass decrease. Is this evolution an artifact? Or, on the contrary, the mass evolves between 850 and 1000ºC. Have the authors performed an analysis of the TG residue to check that the decomposition of the xerogel is completed?

I suggest to provide more information about the thermal analysis characterization, for instance the manufacturer of the TG-DTA SDT Q600 142, the crucibles used in the measurements as well as the typical sample masses. As for the X-Ray diffraction it is important to indicate the voltage and the current of the source.

Language is in general correct but I have found some errors, for instance a missing verb and an incorrect verbal tense in lines 99 and 102 in page 3. In page 4 line 152 “radomle” should be replaced by “randomly”. In page 4 the sentence in line 174 is incorrect. There may be more syntax errors, so a language revision is advisable.

Author Response

The authors are grateful to the Reviewer for such a careful and detailed reading of the manuscript. Below are the responses to comments.

The use of molybdenum carbide as catalyst depends on its porosity. The authors already note that the obtained porosity is relatively low when compared to other synthesis methods. I consider that first a more quantitative comparison with reported results is compulsory; the obtained porosity in this contribution should be compared to those reported in the literature. This relatively low porosity cast doubts about the benefits of using the reported method to obtain molybdenum carbide. Moreover, from Fig 5, the size of molybdenum carbide particles is far above the nanometer range. So given the fact that Nanomaterials journal scope is “involves nanomaterials, with respect to their science and application”, I consider that in its present state this contribution is not suitable for Nanomaterials journal.

This manuscript discusses a new method for synthesis of molybdenum carbide, based on the use of nanoparticles of molybdenum compounds - molybdenum blue. Molybdenum blues are a unique class of compounds in which the dispersed phase is represented by particles of a certain shape and size. Molybdenum blues used in this work are toroidal particles with a predominant size of 3.4 nm. In recent years, it has been widely studied in various fields. And this object belongs to the field of nanomaterials.

Molybdenum blues can be obtained by reducing molybdate solutions. And in this work, an organic substance, ascorbic acid, was used as a reducing agent. This makes it possible to obtain molybdenum carbide by calcining xerogels in an inert atmosphere. Those. the formation of carbide occurs not at the stage of carburization (as in the widespread TPR method), but at the stage of heat treatment, which makes it possible to exclude one stage from the process of obtaining carbide. This significantly reduces the time and simplifies the technological scheme, since there is no need to use flammable gas mixtures (hydrocarbon-H2).

Since molybdenum blue can be obtained in the form of stable dispersions of nanoparticles (sol), the production of materials based on them is a common sol-gel technology. And we can use all the advantages of this method when obtaining supported materials: supported catalysts and catalytic membranes. The preparation and properties of which will be the subject of the next publications of our scientific group (now positive results have already been obtained on the use of the supported and membrane catalysts).

As for the low specific surface area, this is mainly due to the fact that during the preparation of powders during heat treatment, a significant sintering of the primary particles occurs and a significant decrease in the specific surface area occurs. This disadvantage can be avoided if molybdenum blue is deposited on porous supports, which will make it possible to maintain a high dispersion of the supported catalyst. These data are part of the following publication and are not included in this manuscript.

In fig. 5 shows photographs of xerogels obtained by heat treatment at 900ºC. These photographs show rather large objects, but they are aggregates of primary particles of molybdenum carbide, the size of which is 300-400 nm and less, which is a fairly small particle size for the material obtained at high temperatures.

Thermal analysis is an indirect technique, so it is necessary the use of ancillary methods to identify the different steps. In page 7 the authors identify three stages that are assigned to water loss, decomposition of ammonium chloride and formation of molybdenum carbide, but not a single evidence of these assignments is provide. Not even, the observed mass loss is compared to the expected mass loss for such transformations. Therefore, the analysis of the TG curve is quite speculative. The authors should provide evidences of such processes or indicate that their identification is just a guess based on the expected thermal evolution of the precursor.

The authors claim that molybdenum carbide is formed in the temperature range from 750 to 850ºC, but the TG curve on fig. 3 shows that the mass is not stable above 800ºC; there is a smooth mass decrease. Is this evolution an artifact? Or, on the contrary, the mass evolves between 850 and 1000ºC. Have the authors performed an analysis of the TG residue to check that the decomposition of the xerogel is completed?

Analysis of the TG curve shows that 5 regions can be distinguished in which the change in the sample mass occurs at different rates. In the temperature range from the initial value to 100°C, the evaporation of free water retained in the interparticle space occurs. This is indicated by the negative value of the heat effect in this interval. When the temperature rises to 200°C bound water is released. The fact that particles of molybdenum blue are highly hydrated is known in the literature. And the data obtained are in good agreement with them.

 Thermolysis processes occur in the temperature range from 200°C to 700°C. It is known that the breaking of bonds in chemical compounds occurs above 200-250º. For organic substances, thermolysis generally ends at 600-650º. The thermal effects of such reactions are mostly positive. In the presented TGA curve, positive values ​​of the heat effect are also observed in this interval.

In this case, products are released into the gas phase. The analysis of these products was carried out by gas chromatography. It was found that they consist of CO2, H2 and CH4, which are the products of thermolysis of ascorbic acid.

At temperatures of 700-750º there is a sharp change in the heat effect of the reaction, the sample weight and the rate of its decrease, which correspond to the maximum on the DTG curve. Usually, the negative effect of the reaction is attributed to phase transformations or to reactions of formation of substances. Based on the results of XRD data (they were added to the manuscript) we concluded that the formation of Mo carbide occurs in this interval.

Based on the analysis of the products in the gas phase, we found that for the reaction of the formation of molybdenum carbide to occur, the exposure at the final temperature should be 1 hour. Thermal analysis was carried out at a rate of 5°C/min. and during the analysis, the process of carbide formation may not be complete.

Changes have been made to the manuscript of the article concerning the description of the TGA results in section 3.2.

I suggest to provide more information about the thermal analysis characterization, for instance the manufacturer of the TG-DTA SDT Q600 142, the crucibles used in the measurements as well as the typical sample masses. As for the X-Ray diffraction it is important to indicate the voltage and the current of the source.

Additional data of the TGA analysis (manufacturer of the TG-DTA SDT Q600 142, the crucibles) and XRD analysis (voltage and the current of the source) are given in the Methods in section 2.5 (141, 145)

Language is in general correct but I have found some errors, for instance a missing verb and an incorrect verbal tense in lines 99 and 102 in page 3. In page 4 line 152 “radomle” should be replaced by “randomly”. In page 4 the sentence in line 174 is incorrect. There may be more syntax errors, so a language revision is advisable.

The errors mentioned have been fixed, the English language has been improved. Corrections are highlighted in the text of the manuscript.

Round 2

Reviewer 2 Report

The manuscript has clearly improved in the analysis of the TG experimental and with the inclusion of section 3.4, but, the title of the paper misleading: “Synthesis of Nanosized Mo2C by Thermal 2 Decomposition of Molybdenum Blue”. What is in the nanometer range is the precursor, the molybdenum blue, but the size of the molybdenum carbide particles reported is by far above the size of a nanoparticle. I have no doubts that nanoparticles may be obtained when the precursor is dispersed in a nanoporous material, i.e., using a nanostructured template. But this is not what is reported in this paper, the synthetized particles reported in this paper are not nanoparticles. Thus, I consider that the title should be amended as well as the abstract and conclusions.
Minor comments: it is a very relevant information the mass of the samples used in TG experiments. I insist that such an information should be provided in section 2.5, at least a range of masses.
Finally, revise the language in the sentence: “At a temperature 243 of 350 - 400°C, a observed the decomposition of ammonium chloride”.

Author Response

The authors are grateful to the Reviewer for such a careful and detailed reading of the manuscript.

  1. The manuscript has clearly improved in the analysis of the TG experimental and with the inclusion of section 3.4, but, the title of the paper misleading: “Synthesis of Nanosized Mo2C by Thermal 2 Decomposition of Molybdenum Blue”. What is in the nanometer range is the precursor, the molybdenum blue, but the size of the molybdenum carbide particles reported is by far above the size of a nanoparticle. I have no doubts that nanoparticles may be obtained when the precursor is dispersed in a nanoporous material, i.e., using a nanostructured template. But this is not what is reported in this paper, the synthetized particles reported in this paper are not nanoparticles. Thus, I consider that the title should be amended as well as the abstract and conclusions.

The authors agree with the remark about molybdenum carbide nanoparticles. The title, abstract and conclusions were corrected accordingly.

  1. Minor comments: it is a very relevant information the mass of the samples used in TG experiments. I insist that such an information should be provided in section 2.5, at least a range of masses.

Information on the mass of the samples in TGA analysis has been added to section 2.4.

  1. Finally, revise the language in the sentence: “At a temperature 243 of 350 - 400°C, a observed the decomposition of ammonium chloride”.

Sentence was corrected.